# Low-Power, Flexible Sensor Arrays with Solderless Board-to-Board Connectors for Monitoring Soil Deformation and Temperature

**DOI:** 10.3390/s22072814

**Published:** 2022-04-06

**Authors:** Stijn Wielandt, Sebastian Uhlemann, Sylvain Fiolleau, Baptiste Dafflon

**Affiliations:** Earth and Environmental Sciences Area, Lawrence Berkeley National Laboratory, 1 Cyclotron Rd., Berkeley, CA 94720, USA; suhlemann@lbl.gov (S.U.); sfiolleau@lbl.gov (S.F.); bdafflon@lbl.gov (B.D.)

**Keywords:** sensor arrays, geotechnical sensors, solderless connectors, wireless sensor networks, deformation monitoring, accelerometer

## Abstract

Landslides are a global and frequent natural hazard, affecting many communities and infrastructure networks. Technological solutions are needed for long-term, large-scale condition monitoring of infrastructure earthworks or natural slopes. However, current instruments for slope stability monitoring are often costly, require a complex installation process and/or data processing schemes, or have poor resolution. Wireless sensor networks comprising low-power, low-cost sensors have been shown to be a crucial part of landslide early warning systems. Here, we present the development of a novel sensing approach that uses linear arrays of three-axis accelerometers for monitoring changes in sensor inclination, and thus the surrounding soil’s deformation. By combining these deformation measurements with depth-resolved temperature measurements, we can link our data to subsurface thermal–hydrological regimes where relevant. In this research, we present a configuration of cascaded I2C sensors that (i) have ultra-low power consumption and (ii) enable an adjustable probe length. From an electromechanical perspective, we developed a novel board-to-board connection method that enables narrow, semi-flexible sensor arrays and a streamlined assembly process. The low-cost connection method relies on a specific FR4 printed circuit board design that allows board-to-board press fitting without using electromechanical components or solder connections. The sensor assembly is placed in a thin, semi-flexible tube (inner diameter 6.35 mm) that is filled with an epoxy compound. The resulting sensor probe is connected to an AA-battery-powered data logger with wireless connectivity. We characterize the system’s electromechanical properties and investigate the accuracy of deformation measurements. Our experiments, performed with probes up to 1.8 m long, demonstrate long-term connector stability, as well as probe mechanical flexibility. Furthermore, our accuracy analysis indicates that deformation measurements can be performed with a 0.390 mm resolution and a 95% confidence interval of ±0.73 mm per meter of probe length. This research shows the suitability of low-cost accelerometer arrays for distributed soil stability monitoring. In comparison with emerging low-cost measurements of surface displacement, our approach provides depth-resolved deformation, which can inform about shallow sliding surfaces.

## 1. Introduction

Landslides are a global and frequent natural hazard affecting many communities and infrastructure networks. Between 2004 and 2016, 4862 nonseismic fatal landslide events were recorded in the Global Fatal Landslide Database [1]. Of those, 12% had a direct impact on infrastructure; this is most likely an under-reported value [1,2]. Small failures (<10 m${}^{3}$) are recognized to have the largest impact on infrastructure, representing 95% of events that impacted the Swiss transportation network between 2012 and 2016 [3]. While these events account for 74% of direct damages, they are rarely reported in regional or global landslide inventories. Climate change is expected to increase the frequency of extreme weather events, which are known to have adverse impacts on infrastructure slopes [4], and thus may cause even more failures in the future. Hence, there is an increasing need for novel instrumentation that can lead to a better understanding and enhanced monitoring of infrastructure conditions [5].

Slope instabilities, i.e., landslides, can have various forms and triggering processes [6]. Most landslides are triggered by two factors: earthquakes or extreme rainfall events. Rainfall-induced landslides are often caused by a rise in pore pressure, and hence are related to variations in soil moisture content. Here, we focus on these types of landslides, which can occur both in natural and engineered slopes (such as infrastructure slopes or tailings dams).

Smethurst et al. and Uhlemann et al. [5,7] provide a comprehensive overview of current technological solutions for slope stability monitoring. Their studies highlight the frequent use of technologies to monitor soil displacements but also emphasize the need for depth-resolved, continuous measurement approaches. While slopes with known stability problems are often monitored using conventional instrumentation, such as inclinometers or piezometers and/or site-specific geophysical instrumentation or soil moisture sensor networks [8,9], such efforts are often related to significant costs for sensor installation and long-term monitoring. Even though such site-specific monitoring solutions are known to be valuable for managing infrastructure assets [10], stakeholders require new approaches for monitoring the conditions of longer lengths of earthworks [5]. While remote sensing or fiber optic sensing based solutions become increasingly available, they are often very costly, require complex installation and/or data processing schemes, and lack vertical resolution. Remote sensing data are highly sensitive to ground surface deformation (i.e., InSAR) and are shown to be successful in imaging deformation of failing infrastructure [11,12], but the temporal resolution is poor and the spatial resolution often not sufficient to detail failure events [11]. Fiber-optic sensing techniques, such as distributed strain or acoustic sensing (DSS and DAS, respectively), are becoming increasingly used to monitor infrastructure assets [13]. Since telecommunication fibers are often installed next to road or railways, these networks of fiber-optic cables can provide a cost-efficient monitoring solution [14]. Yet, the interrogators required to record and analyze the signals are expensive and need a reliable power source. Particularly for remote sensing techniques, data processing is computationally expensive, limiting their use for real-time assessment of slope conditions [13]. The monitoring system presented in this paper addresses the shortcomings of fiber-optic and remote sensing techniques by providing a low-cost monitoring solution with spatiotemporally dense, vertically resolved data sets. Furthermore, the system features low-complexity installation and data processing.

Wireless sensor networks (WSNs) comprising low-power, low-cost sensors [15] can overcome some of those limitations. WSNs are now used as part of landslide early warning systems [16], highlighting their use for rapid assessment of slope conditions. However, WSNs, with their discrete point measurements, may not fully capture the spatial heterogeneity of deformation dynamics, but they can be linked to observations from remote sensing or other ground-based sensing techniques [17]. Furthermore, the use of densely distributed sensor networks in remote field conditions poses many challenges in terms of power provisioning, cost, and network reliability, particularly during critical events where data transmission may be limited due to environmental conditions. Long-range network protocols with dedicated communication schemes can enable wireless connectivity in the field [18], while cost and power concerns can be addressed with custom hardware designs.

Here, we present a novel low-power, low-cost sensing approach for soil deformation monitoring. In combination with low-power, long-range wireless connectivity [19], the proposed solution becomes a means for monitoring the stability of infrastructure earthworks, or natural slopes. Our solution is based on three-axis accelerometers, which are deployed as a linear array of sensors in a flexible probe. The accelerometers regularly measure the probe’s deformation, but measurements are not performed continuously in order to minimize power consumption. This means that a probe regularly checks for a change in static conditions.This approach is comparable with commercially available ShapeAccelArrays (SAA, Measurand, Hanwell, New Brunswick, Canada [20]) and the prototype for inclination measurements presented by Ruzza et al. [21]. Here, we develop an integrated solution that houses sensors in a thin tube. The sensor probe is used with a low-cost logging unit that is described in detail in Wielandt et al. [22], resulting in a solution that has a small footprint, is sturdy and easy to install, is highly energy efficient (i.e., AA battery powered), and ready to be deployed as networks of sensors [18]. By combining these deformation measurements with depth-resolved temperature measurements, we can link our data to water and temperature dynamics [23,24,25]. Water dynamics control pore pressures in the subsurface and, hence, often trigger slope instabilities [26]. Recent work has shown that soil water content and/or soil physical properties can be derived from distributed temperature measurements [27,28,29,30], and hence, combining displacement measurements with those distributed temperature measurements will eventually enable us to not only detect slope failure but also its cause. Similarly, in permafrost regions, permafrost thaw is known to cause slope instabilities [31]. Having a good understanding of the temperature evolution with depth enables us to further study the link between these two variables: temperature and soil deformation. In comparison with emerging, low-cost measurements of surface displacement (e.g., tiltmeter, extensiometer or GPS), our approach provides depth-resolved deformation, which can inform about shallow sliding surfaces and processes.

The development of these linear sensor arrays poses many challenges. From an electrical point of view, it requires the design of a low-power array of individually addressable sensors, with a configurable length of up to ∼2 m. In order to enable large-scale distributed wireless sensor deployments, the design should also be low cost, and easy to manufacture and assemble [32]. This means that the electrical challenges are complicated by mechanical requirements and limitations that mandate specific design solutions and novel techniques. Critical mechanical requirements include a thin and flexible probe design with a fixed 50 mm spacing between sensors and robustness in harsh field conditions. In order to fulfill all requirements, a modular approach is adopted. Sensor sections measuring 200 mm long are designed on semi-rigid, narrow, FR4 printed circuit boards (PCBs). These boards are cascaded by means of a novel, solderless board-to-board connection technique [33]. In contrast with existing (commercial) solutions, the proposed technique has a zero-cost impact on board manufacturing and provides great mechanical robustness at a minimal cross-section, enabling 5 mm thin sensor arrays that can be placed in a thin, semiflexible tube. The connector assembly is fully designed in PCB technology, does not require soldering, and consists of a male part containing metal-plated barbed pins and a female part containing metal-plated slots. After cascading the boards and inserting them into the plastic tube, the tube is filled with an epoxy compound to withstand field conditions.

The electromechanical properties of the proposed design are experimentally evaluated and presented in this paper. We discuss the resistance for different signal traces and evaluate the change in contact resistance over a one-year time period. Furthermore, we assess these electrical parameters as a result of epoxying, heating, freezing, and bending the probe. Besides these tests that focus on the connector design, we study the probe’s power consumption for different configurations, and we perform a mechanical evaluation of the probe’s flexibility. Furthermore, we assess the accuracy of the probe for deformation measurements. Lab tests demonstrate the probe’s ability to accurately detect soil movements. Finally, the probe operation is evaluated in an arctic environment, illustrating that the combination of temperature sensors and accelerometers can be used to study soil movements in permafrost environments.

## 2. Materials and Methods

### 2.1. Inclination Measurements Using Accelerometers

Accelerometers are microelectromechanical systems (MEMS) that measure a device’s acceleration along—usually—three axes. The developments in consumer electronics over the past decade have resulted in a competitive offering of low-cost, high-performance accelerometer chips. In conditions with low to zero acceleration, these sensors can be used for discontinuous inclination sensing by measuring Earth’s gravity along three axes [34]. The intention is to regularly measure soil deformation and not to continuously monitor the ambient seismicity. This purposeful limitation results in a low-power system that handles regular, small amounts of data.

Figure 1a depicts a static configuration with gravitational acceleration vector g applied to a tilted accelerometer. This acceleration of $9.81\;m\;\cdot \;s^{-2}$ is referred to as 1 g throughout the remainder of this paper. A reference coordinate system (xyz) can be aligned with g, such that the *z*-axis opposes g. Any applied acceleration is measured by the accelerometer as a vector $a=[a_{x},a_{y},a_{z}]$. Under perfect alignment with the reference system, $a_{z}=-1$ g, but because of the inclination of the sensor, both coordinate systems are rotated by θ, ψ, and ϕ. As illustrated in Figure 1a, these angles express the rotation between $a_{x}$ and the xy plane, $a_{y}$ and the xy plane, and $a_{z}$ and *z*, respectively. Basic trigonometry leads to Equations (Equation 1)–(Equation 3), which can be used to calculate θ, ψ, and ϕ, from $a_{x}$, $a_{y}$, and $a_{z}$ [35].
(1)$$\begin{array}{c} \theta =\tan^{-1}\left(\frac{a_{x}}{\sqrt{a_{y}^{2}+a_{z}^{2}}}\right) \end{array}$$
(2)$$\begin{array}{c} \psi =\tan^{-1}\left(\frac{a_{y}}{\sqrt{a_{x}^{2}+a_{z}^{2}}}\right) \end{array}$$
(3)$$\begin{array}{c} \phi =\tan^{-1}\left(\frac{\sqrt{a_{y}^{2}+a_{x}^{2}}}{a_{z}}\right) \end{array}$$

The accuracy of an inclination measurement is affected by a number of error sources, including noise, drift, and offset [36]. Bad noise performance is usually caused by mechanical vibrations, high measurement bandwidth, and instability of the supply voltage. For soil deformation sensing, the expected impact of noise is minimal because the outdoor environments of interest are usually vibration free, measurement bandwidths can be kept low, and supply voltage can be buffered. In infrastructure-monitoring scenarios (e.g., railways and highways), vibrations might be more prevalent. In this case, sensor noise can be reduced by averaging measurements over time and lowering the sensor’s measurement bandwidth below the vibrational frequencies. Sensor drift can happen over time and under influence of changes in temperature and/or supply voltage. For subsurface applications, temperature changes are usually limited and slow. Furthermore, drift can be detected, compensated, and prevented by stabilizing the supply voltage and by performing measurements of temperature and battery voltage. Offset errors (due to the absolute temperature or the sensor’s manufacturing process) are usually eliminated through a calibration process. Deformation measurements, however, are not affected by offsets and do not require calibration, since only changes in inclination are measured.

### 2.2. Sensor Arrays for Deformation Monitoring

Figure 1b presents an array of accelerometers ($a_{i}$, i∈{1,…,n}) for soil deformation sensing. The fixed 100 mm spacing between accelerometers enables a precise assessment of the deformation of the probe over time. As depicted in Figure 1b, the accelerometer chips are positioned in a spiraling formation around the probe’s axis (which is usually aligned with the *z*-axis) with a $90^{{}^{\circ}}$ step size. The benefit of this formation is two fold. First, in practice, the accuracy of acceleration measurements often varies by axis [37], so if all accelerometers were mounted in the same plane, inaccuracies would accumulate anisotropically. This can be overcome by the $90^{{}^{\circ}}$ spiraling arrangement of the accelerometers’ coordinate systems around the *z*-axis, which introduces an alternation of measurement axes along the probe. For example, a probe tilt in the zy plane in Figure 1b would be perceived along $a_{2,z}$, $a_{4,y}$, $-a_{6,z}$, $-a_{8,y}$, …, effectively evening out differences in inaccuracy between the $a_{z}$ and $a_{y}$ measurements. Secondly, the proposed arrangement prevents any asymmetrical sensor drift from accumulating along the sensor array, whether it be time, voltage, or temperature induced. The probe is also equipped with temperature sensors ($T_{i}$) spaced at a 50 mm distance from the accelerometers. These temperature measurements can be used for the detection and/or correction of temperature effects on the accelerometers, and in some cases provide valuable information on subsurface processes, including water and temperature dynamics [29].

To obtain the global deformation along a probe, all accelerometer coordinate systems have to be aligned along a common direction. Furthermore, the orientation of the entire probe can be registered during deployment, but this is usually not necessary for landslide monitoring since the deformation occurs in the direction of the slope. Here, we align all sensors with the bottom sensor $a_{n}$. However, the practical realization of a cascaded sensor formation might not always follow a perfect rotational movement in $90^{{}^{\circ}}$ steps. In order to correct for the rotational movement and its potential errors, we use Rodrigues’ rotation formula for three-dimensional vector rotations. The aligned version $\left(r_{i}\right)$ of the acceleration vector $a_{i}$ is calculated following Equation (Equation 4), and the rotation matrix (${\mathrm{Rot}}_{i}$) is computed following Equation (Equation 5). ${\mathrm{Rot}}_{i}$ is determined in a calibration phase before deployment by placing the probe under a $∼45^{{}^{\circ}}$ angle. This assures that $a_{x}$, $a_{y}$, and $a_{z}$ are nonzero for all sensors.    
(4)$$\begin{array}{c} r_{i}={\mathrm{Rot}}_{i}\;\cdot \;a_{i} \end{array}$$
(5)$$\begin{array}{c} {\mathrm{Rot}}_{i}=I+K_{i}\sin\gamma_{i}+\left(1-\cos\gamma_{i}\right)K_{i}^{2} \end{array}$$

*I* is the 3 × 3 identity matrix, and $\gamma_{i}$ is the angle between vector $a_{n}$ and $a_{i}$ as expressed in Equation (Equation 6). $k_{i}$ is the unit vector that defines the axis of rotation in Equation (Equation 7), and $K_{i}$ is its cross-product matrix as defined in Equation (Equation 8).
(6)$$\begin{array}{c} \gamma_{i}=\frac{a_{n,x}\;\cdot \;a_{i,x}+a_{n,y}\;\cdot \;a_{i,y}+a_{n,z}\;\cdot \;a_{i,z}}{\sqrt{a_{n,x}^{2}+a_{n,y}^{2}+a_{n,z}^{2}}\sqrt{a_{i,x}^{2}+a_{i,y}^{2}+a_{i,z}^{2}}} \end{array}$$
(7)$$\begin{array}{c} k_{i}=\frac{a_{n}\times a_{i}}{|a_{n}\left|\right|a_{i}|\sin\gamma_{i}} \end{array}$$
(8)$$\begin{array}{c} K_{i}=\left[\begin{array}{ccc} 0 & -k_{i,z} & k_{i,y}\\ k_{i,z} & 0 & -k_{i,x}\\ -k_{i,y} & k_{i,x} & 0 \end{array}\right] \end{array}$$

After aligning all accelerometers, the relative displacement for each sensor $d_{i}=\left[d_{i,x},d_{i,y},d_{i,z}\right]$ is determined with the bottom accelerometer as a reference. The displacement vector $d_{i}$ is calculated following Equations (Equation 9)–(Equation 11), which accumulate the displacements between successive sensors starting at the bottom of the probe. These relative displacements between neighboring sensors are defined by the inclination of the sensor (θ, ψ, and ϕ, according to Equations (Equation 1)–(Equation 3), respectively), multiplied by their intersensor distance of 100 mm.
(9)$$\begin{array}{c} d_{i,x}=\sum_{j=n-1}^{i}\sin\theta_{i}\;\cdot \;100\;\left[\mathrm{mm}\right] \end{array}$$
(10)$$\begin{array}{c} d_{i,y}=\sum_{j=n-1}^{i}\sin\phi_{i}\;\cdot \;100\;\left[\mathrm{mm}\right] \end{array}$$
(11)$$\begin{array}{c} d_{i,z}=\sum_{j=n-1}^{i}\sin\psi_{i}\;\cdot \;100\;\left[\mathrm{mm}\right] \end{array}$$

### 2.3. Electronic Design

In order to build the low-cost, flexible sensor probe envisioned in Figure 1b, an electronic circuit with cascaded sensors was developed. Figure 2 presents a low-complexity design that enables narrow, flexible board configurations and resembles the thin temperature probe design presented in [32]. The use of digital sensors with two-wire interface (TWI, also known as I${}^{2}$C) and address pins enables communication with multiple identical sensors on a single communications bus. In order to individually address the sensors, each sensor’s address input is connected to a D flip-flop’s output, as presented in the schematic. The configuration of D flip-flops effectively forms a shift register with each bit being connected to a separate temperature sensor/accelerometer pair. Using the ‘D’ and ‘CLK’ signals, a single address bit can be propagated along the entire sensor array. This approach allows cascaded sensor configurations of arbitrary length using just six signals, including the I${}^{2}$C signals (‘SCL’ and ‘SDA’), power supply ‘VCC’, and ground ‘GND’.

All selected components are compatible with the 2.0 V to 3.6 V range, which allows operation on a 3 V lithium battery or 2× AA batteries without voltage conversion circuitry, preventing energy loss [22]. The focus on low-power components maximizes the battery life of the connected data logger, but it also ensures voltage stability along the probe. Power supply noise is further reduced with RC-filters at every sensor’s power inputs, ensuring a minimal impact on the measurements. Taking the electrical requirements into account, SN74LVC1G175DCK [38] was selected as a low-cost, low-power, discrete, small-footprint D flip-flop. Temperature measurements are performed with the TMP117AIDRVR [39], which satisfies the same electrical requirements and provides an extremely high resolution of $0.0078125{\;}^{{}^{\circ}}C$ and a factory-assured accuracy of $\pm 0.1{\;}^{{}^{\circ}}C$ in the −20–50 ${}^{{}^{\circ}}C$ range. The field of commercially available accelerometers is extensive, but the wide set of requirements led to the selection of the ADXL345 from Analog Devices [37]. This chip provides a measurement resolution of 3.9 mg, which results in a displacement resolution of 0.39 mm under 100 mm sensor spacing. It is offered at a low price point, provides acceleration measurements in the ±2 g range, has an LGA-14 footprint, and a pin layout that facilitates its use on narrow circuit boards. Furthermore, the chip satisfies the previously formulated electrical requirements, which include compatibility with I${}^{2}$C and the presence of an address pin. The available documentation [35,36,37] provides a detailed description of the chip’s error sources, including noise, drift, and offset. As previously discussed, the offset does not affect deformation measurements. However, temperature and supply-voltage-induced drift should be minimized, detected, and/or corrected. Temperature fluctuations are naturally minimized by the probe’s deployment in the subsurface, while voltage fluctuations are minimized by RC-filters and the use of batteries with a stable output voltage independent of charge state and temperature (e.g., Li/FeS${}_{2}$ Energizer L91 [40]). In order to enable the detection and correction of any occurring drift, each measurement with the deformation probe is accompanied by a soil temperature profile and battery voltage measurement. Sensor noise is minimized by a combination of measures; (i) the measurement bandwidth of the accelerometer is set to 25 Hz, which provides the lowest noise level [37] and a low supply current of 90μA, and (ii) averaging of k∈{1,2,4,8,16,32} measurements. We investigated the impact of averaging on the probe’s power consumption and accuracy of the deformation measurements.

#### Device Integration and Power Consumption

The developed sensor probe is intended for long-term, distributed soil temperature and deformation monitoring. This means that the probe should be connected to a low-power, battery-operated, low-cost data logger that provides local data storage and/or wireless connectivity, time keeping provisions, and limited data processing. For this research, we employed the logger presented in [22], which is equipped with an NRF52832 ARM Cortex M4, 32 Mb low-power NOR flash memory, a microSDHC expansion slot, a PCF2129AT real-time clock, and wireless connectivity through Bluetooth Low Energy (BLE) and LoRa. The BLE interface enables researchers to manually download data to their smartphone in the field. The LoRa interface provides real-time data transmission over several kilometers (depending on the environment [41]) to a base station that is connected to the internet. The logger connects to the probe through a six-wire interface and uses a TPS22919 load switch to cut off the probe’s power supply when no measurements are performed. The I${}^{2}$C bus is operated at 50 kHz with a TCA9803 bus buffer with integrated current sources, guaranteeing signal integrity for probes of up to at least 2 m in length. The process of initializing, reading, and averaging sensor data is shown in Algorithm 1. The logger operates in the 1.8–3.6 V range, which makes it compatible with the proposed 2× AA battery set of Energizer L91 Li/FeS${}_{2}$ cells with a capacity of 3500 mAh in the temperature range of $-40{\;}^{{}^{\circ}}$C up to $60{\;}^{{}^{\circ}}$C [40]. The logger’s average power consumption equals 75.9μW, not taking the sensor probe into account.
**Algorithm 1:** Probe Measurement1:Turn on probe power supply2:Wait 25 ms for signals to settle and devices to power up3:**for** 
Sensor=1,2,…,n 
**do**4:    Initialize temperature sensor and accelerometer and start measurements5:**end for**6:**for** 
$step_{avg}=1,\ldots ,k$ 
**do**7:    Wait for measurements to complete8:    **for** Sensor=1,2,…,n **do**9:        Read new sensor values10:    **end for**11:    **if** $step_{avg}<k$ **then**12:        Restart accelerometer measurement13:    **end if**14:**end for**15:Turn off probe power supply16:Average *k* accelerometer measurements

The characterization of the probe’s power consumption is critical to assess its impact on the end device’s battery life. We used a Keithley DMM6500 6.5 digit digital multimeter to measure the current along the VCC line under a constant voltage of 3.3765 V. The current measurements were performed with an integration interval of 1.16 ms and a 10× averaging filter, resulting in a total data acquisition rate of 86 samples per second. We evaluated the power consumption for different configurations, varying the number of sensors (*n*) and the number of averaged measurements (*k*). As such, we obtained power profiles over time for the evaluated configurations. The total energy consumption for an entire probe measurement is calculated by integrating these power profiles over time.

### 2.4. Electromechanical Design

The envisioned concept results in a set of challenging mechanical requirements that affect the electrical design of the probe. The probe should consist of a thin, low-cost, weather-resistant tube that is sufficiently flexible to follow soil movements but rigid enough to allow the probe to be pushed into a tight hole in the ground. The sensor array schematically depicted in Figure 2 should maintain an accurate spacing of 50 mm between sensors and enable its insertion into a flexible tube. This means that this electrical setup should be rigid enough to provide fixed sensor spacing but flexible enough to withstand the deformation of the probe, as well as thermal expansion. Furthermore, manufacturability, cost, and production yield should be considered to facilitate large-scale, distributed deployments.

For the construction of the sensor array, several technological options exist. The components can be mounted on flexible circuit boards, but such setup would not provide the rigidity required for fixed sensor spacing and spiraling sensor configurations [42]. Thin, fiber-based (FR4) circuit boards provide a low-cost, semiflexible solution that enables fixed sensor spacing [43]. Because PCBs cannot be manufactured for the entire probe length, we adopted a modular approach with each 20 cm module containing two temperature sensors and two accelerometers. The resulting setup of thin, cascaded, FR4 circuit board sections provides a low-cost solution for various probe lengths and enables the spiraling configuration presented in Figure 1b. However, the electrical and mechanical connections between the cascaded boards poses another design challenge. Many connectivity solutions have been available for decades, but despite extensive evaluations of commercial technologies, no adequate connector solution was found. The summary below indicates the shortcomings of existing solutions:Connector–cable–connector setups: This labor intensive, costly, and bulky solution is often found in prototypes and low-volume production devices [44]. Each sensor board has an incoming and outgoing connector soldered onto it, and all boards are connected with cable assemblies. The cost of the connector terminals, cable assemblies, soldering, and manual assembly easily exceeds the actual cost of the sensors.Direct PCB-to-PCB soldering: This technique is often found in cascaded LED strips and constitutes PCB edge pads that are aligned and soldered together. This approach is labor intensive, prone to production errors, and often mechanically unreliable, as mechanical stress can result in solder cracks [45].Board-to-board connectors: Solder-mounted board-to-board connectors are commonly used in electronic designs and can accommodate the required fixed sensor spacing when mounted at the edge of the board. However, these parts are usually costly, affect production yields and cost, are often unreliable under mechanical stress, and—most importantly—require too much board space, affecting the thickness of the entire probe [44].

Because of these disadvantages, a novel board-to-board connection technique was developed. The proposed approach does not require any components and does not require soldering. The technique relies on two complementary PCB designs that can be classified as female (Figure 3a) and male (Figure 3b). The main operating principle is that these two FR4 PCBs snap into each other as demonstrated in Figure 3c, with the metalized areas of the boards being pressed together, creating reliable electric contacts.

The male connector consists of 0.8 mm thick FR4 board with a design that features an array of six 3.0 mm wide pins. These pins have exposed metalization (electroless nickel/immersion gold) on the top and bottom layer of the PCB and fit into the female part. The female connector contains a series of through-plated PCB slots of 0.9 mm × 3.2 mm, angled under $\pm 12^{{}^{\circ}}$. The angles of the PCB slots introduce a degree of spring loading that ensures reliable electrical contact between the pins and the plated slots. Furthermore, the barbed pin design in the male connector allows both parts to lock together. The final solution is a connector design that is entirely made in PCB technology using standard, low-cost production methods. As depicted in Figure 3c, the boards are mated under $90^{{}^{\circ}}$ angles, realizing the spiraling sensor formation that was presented in Figure 1b. The assembly of a connector can be performed in fewer than two seconds and does not require soldering. This solderless technique contributes to the reliability of the connector under mechanical stress, similar to existing—yet costly—press-fit connectors [44]. The total width of the boards is only 5.5 mm, so the sensor array can be placed in a thin, flexible tube. At the top of the probe, a wire-to-board adapter (illustrated in Figure 3d) provides connectivity to the data logger through a 1 mm pitch 6-pin JST SH connector [46].

The sensor boards exhibit a particular PCB design with narrow (1.60 mm) sections that connect the sensors and connectors at the board ends (Figure 4). These narrow board sections provide flexibility in multiple directions and reduce stress on electronic components and their solder joints. Other provisions to handle mechanical stress include fillets of the board outline around the sensors and PCB trace tapering and curving instead of using $45^{{}^{\circ}}$ turns. One can also observe that the narrow interconnecting sections are designed in a zig-zag fashion. This implementation introduces a minimal amount of elasticity along the sensor array that relieves mechanical stress due to thermal expansion of the boards. The temperature at which the probes are assembled and filled with epoxy can easily be $50{\;}^{{}^{\circ}}C$ above the use-case temperatures (e.g., in the Arctic). According to [47], a thermal expansion coefficient of 15.30 ppm/${}^{{}^{\circ}}$C can be expected along the length of an FR4 board. This means that a $50{\;}^{{}^{\circ}}C$ temperature drop would result in a 1.377 mm crimp along a 1.8 m long probe.

The spiraling sensor assembly is inserted in a plastic tube of 9.5 mm outer diameter and 6.35 mm inner diameter. We selected cellulose-acetate-butyrate tubing because of its low cost, weather resistance, flexibility, and structural stability over a wide temperature range. Finally, the tube is filled with the Epoxies Inc. (Cranston, RI, USA) urethane blend 20-2360 [48], which was also selected for its structural stability over a wide temperature range.

#### 2.4.1. Probe Flexibility Tests

To evaluate the developed design, a series of mechanical and electrical tests were performed. First, we evaluated the flexibility of the probe. Figure 5 represents the test setup for this experiment: a probe is anchored and a force F is applied 200 mm from the anchor point, causing a deformation $d_{F}$. The force and deformation are measured in a horizontal plane to rule out any gravitational impact on $d_{F}$. This experiment is performed in two configurations, taking into account the orientation of the sensor board in the studied 200 mm probe section. In configuration (a) F is applied in the PCB plane, whereas (b) studies the deformation for a force perpendicular to the PCB plane.

#### 2.4.2. Connector Evaluation

Since the developed connectors constitute a novel technology, multiple tests and evaluations were performed to evaluate the stability, consistency, and reliability of the electrical contacts. For all measurements of contact resistance, we used the Keithley DMM6500 (Beaverton, OR, USA) digital multimeter in a 4-wire configuration to eliminate any influence of measurement leads. All measurements were performed at room temperature unless noted otherwise.

In a first study, we evaluate the resistance of the GND, VCC, SDA, SCL, and CLK traces along two 1.2 m probes. These trace resistances are measured separately for each of the 12,200 mm probe sections without taking the contact resistance of the board-to-board connectors into account. Next, the trace resistances are measured for the entire length of the probes, which includes all 12 connectors. Aarts et al. (2008) [49] describe a very similar approach, evaluating contact resistance of several daisy-chained interconnects. The measurements allow us to evaluate the average trace resistance per sensor board and assess the impact of the connectors’ contact resistance.

The next study evaluates the impact of different events on the trace resistances. For these tests, we find the average change in trace resistance (Δ*R*) per 200 mm probe section by—again—measuring the change over an entire 1.2 m probe. In a first test, we measure the change in resistance after filling the probe with the epoxy compound. Next, the probe is frozen for 6 h at $-23.7{\;}^{{}^{\circ}}C$, and the resistances are measured in the first 2 min of the defrosting process, as well as after temperature equilibration. In a different test, a probe is placed in an oven at $103{\;}^{{}^{\circ}}C$ to $108{\;}^{{}^{\circ}}C$ for six hours. For safety purposes, no resistance measurements were performed on the hot probe, but the resistances were reassessed when the probe was back at room temperature.

The next experiment focuses on connector reliability under mechanical stress. This means that all resistance measurements are performed while bending the probe. We evaluate the average Δ*R* per 200 mm probe section again by performing measurements on a 1.2 m probe, as a function of the probe bending radius.

The last connector experiment evaluates contact resistance over time. For these tests, we evaluate the average Δ*R* per 200 mm probe section again by performing resistance measurements on 1.2 m probes. We performed these measurements over the course of one year to analyze the effects of oxidation and creep in the connectors.

### 2.5. Functional Probe Evaluation and Accuracy Assessment

To estimate the total uncertainty of the probe’s deformation, different parameters have to be taken into account, including the sensors’ measurement errors, the number of averaged measurements (*k*), the probe design ($90^{{}^{\circ}}$ rotation of each segment), and the length of the probe. In order to evaluate the accuracy of the calculated deformation as a function of the number of averaged measurements (*k*) for a 1 m probe, 3000 acceleration samples were acquired in a stable environment (0 mm displacement) at constant temperature. Then, a Monte Carlo simulation of 10,000 iterations was performed, randomly combining measured samples to create 10,000 virtual probes of 1 m in length. For each sensor, the displacement in *x* and *y* direction was calculated following Equations (Equation 9) and (Equation 10), respectively. Finally, the deformation error ($\epsilon_{d}$) was evaluated for a confidence interval of 95% (percentile 2.5 and 97.5) considering the relative displacement between the first and last sensor.

The temperature sensors are used to characterize shallow subsurface processes (e.g., permafrost thaw, variations in soil moisture), but temperature values can also provide insights in the accuracy of deformation measurements. As stated by the manufacturer, temperature changes can induce a 0 g offset of acceleration measurements. We evaluated this effect on $\epsilon_{d}$ for $\pm 1{\;}^{{}^{\circ}}$C and $\pm 10{\;}^{{}^{\circ}}$C temperature fluctuations, relying on the performance characteristics reported in the ADXL345 datasheet [37].

In order to evaluate the operation of the probe in a real-world scenario, and to better understand the impact of permafrost degradation on soil and carbon transport, 60 probes (1.2 m to 1.8 m long) were installed in September 2021 at a field site on the Seward Peninsula, AK. All data was collected with a smartphone in the field using the logger’s BLE interface because the lack of LoRa base stations and internet connectivity in the studied field site prevents real-time data transmissions. We focus on a single probe to evaluate its imaging performance of small deformations occurring over a five-day time window. Because of its location in a permafrost environment, where changes in temperature may cause soil displacements, this also demonstrates the importance of jointly measuring temperature and deformation.

## 3. Results

### 3.1. Power Consumption

Figure 6 presents the power profile of two probe configurations with different lengths and a different number of averaged measurements (*k*). The plots indicate the different phases of measurement Algorithm 1: an initial peak of the power consumption occurs at power-up and is followed by a wider peak that can be associated with sensor initialization and measurements of temperature and acceleration values. After this initial measurement, a series of narrow power peaks represents the remaining k−1 acceleration measurements. This explains the longer duration of the k=32 measurement with respect to the k=8 configuration. It should also be noted that a probe measurement with n=12 and k=32 takes almost 2.5 s, which is significantly longer than the time needed to perform 32 measurements at the selected 50 Hz output rate. This phenomenon can be attributed to the relatively slow I${}^{2}$C bus CLK signal of 50 kHz and the amount of sensor data that is exchanged. As expected, the length of the probe clearly affects its average power consumption. For the 1.2 m and 1.8 m probe, we measured an average power consumption of respectively 6.17 mW and 13.20 mW, which corresponds to an average supply current of 1.83 mA and 3.91 mA. One could argue that these values allow the entire probe to be powered through a microcontroller’s GPIO pin and that the load switch on the data logger is not strictly necessary to switch the probe on or off. However, a load switch does accommodate for larger current spikes without directly affecting the power supply of the microcontroller or sensors. Cutting off the power supply of the probe with the selected TPS22919 load switch contributes to the desired low-power operation of the probe: the probe’s supply current between measurements is below the 1 nA detection limit of the used multimeter.

To assess the impact of the probe’s power consumption on the battery life of a device, we evaluate the required energy per probe measurement, as presented in Figure 7. These results demonstrate that the required energy increases more than linearly as the length of a probe increases. This is explained by the impact of probe length on both measurement duration and power consumption. When considering the impact of *k* on the energy per measurement, one can observe that both parameters do not scale proportionally due to the energy cost of power-up, initialization, and temperature measurements. For the evaluation of battery life, we investigate a worst case battery life scenario of k=32, n=18, and a probe measurement interval of 15 min. Taking into account the data logger’s average sleep power consumption of 75.9 μW, and the logging energy of 4.747 mJ [22], we can calculate the daily energy budget of 7.01 J for the data logger, and 2.84 J for the sensor probe. These minimal power needs allow long-term monitoring solutions using small batteries or solar panels. For the proposed 2× AA 3500 mAh battery set, a theoretical battery life of 4320 days (almost 12 years) was calculated. However, for a more realistic battery life estimation, self-discharge and fluctuations in power consumption should be taken into account. Since the sensor probe represents only 29% of the device’s total energy consumption in this worst case scenario, sacrifices in probe length or averaging (*k*) would only result in insignificant battery life improvements.

### 3.2. Probe Flexibility

Figure 8 presents the deformation $d_{F}$ of the probe by applying a force either (a) in the PCB plane or (b) perpendicular to it. A small force of ∼0.2 N results in a considerable deformation (2.4 mm or 2.8 mm) of the studied 200 mm probe section. Increasing this force to 10 N results in an extreme deformation of the probe, but $d_{F}$ does not vary linearly with F. Scenario (a) and (b) exhibit similar results, so the flexibility of the probe can be considered nearly isotropic. In scenario (a), a <1 N force results in a slightly larger deformation than in scenario (b), which contrasts expectations. The logarithmic scale amplifies the apparent significance, but the absolute differences of these $d_{F}$ values are all ≤0.7 mm and could be caused by measurement inaccuracies.

To monitor slope instabilities, the applied forces due to soil movements should exceed the probe’s resistance to deformation. However, accurate calculations of these interactions involve complex models that take into account soil and slope characteristics, as well as the physical parameters of the probe. Such calculations exceed the scope of this research, but a basic numerical approach can provide sufficient insight in the order of magnitude of the forces that can be expected. Das et al. (2012) [50] provide an example for the calculation of stress in a soil mass, assuming standard soil characteristics and a slope angle of $20^{{}^{\circ}}$, resulting in a shear stress of 111.5 kN/m${}^{2}$. The approximate probe cross-section of 1 cm${}^{2}$ results in a 11.15 N force. Given the extreme deformation observed under 10 N, one can conclude that the probe does not resist soil movements and that the deformation of the probe is representative for the soil movement.

### 3.3. Electromechanical Connector Characteristics

To evaluate the reliability of the developed connector solution and assess its performance in various conditions, we present the results of a contact and trace resistance study. This includes changes in resistance as a function of time, temperature, and probe deformation.

#### 3.3.1. Trace Resistance

Figure 9 presents the resistance of each signal trace along a 200 mm board section, with and without the connectors’ contact resistance. First, we calculate an ideal theoretical scenario without interconnects. For this scenario, we approach the resistance of each copper PCB trace by assuming a length of 200 mm, a thickness of 35 μm (which is a manufacturing parameter), and a resistivity $\rho_{20}\;=\;1.678\;\times \;10^{-8}\;$Ωm at 20${\;}^{{}^{\circ}}$C [51]. All traces have a 0.2 mm width, except for the ground plane, which is close to 0.4 mm width in most places. The resulting theoretical trace resistances are, respectively, 0.479Ω and 0.240Ω per 200 mm probe section, and are visualized in Figure 9. The individual trace resistances (without connectors) for 200 mm board sections are spread closely around the theoretical values. GND trace resistance is slightly lower than the theoretical value, which can be attributed to the use of irregularly shaped ground planes that were theoretically approached by a 0.4 mm wide trace. Outlying trace resistance values for the same board were either labeled as ‘A’ or ‘B’ in Figure 9. The measurements for board ‘A’ consistently show a low resistance for each trace, which is most likely attributed to a board manufacturing variation (e.g., thicker copper layer). Outliers for board ‘B’ are less pronounced (<6% above average) and limited to the VCC and SCL trace. Since these traces are located on the bottom PCB layer, one could assume a single-sided board manufacturing variation. When observing the average trace resistance with and without connector, it is clear that the impact of connector resistance is barely noticeable: the average increase in trace resistance is only 1 mΩ. Along a 1.8 m probe, this would lead to an average increase in contact resistance of 9 mΩ, which is negligible compared with the total PCB trace resistance. The connector resistance is also low in comparison with the used 1 mm pitch 6-pin JST SH connector at the end of the probe, which features an initial contact resistance of up to 20 mΩ [46]. In general, the average trace resistance aligns well with the expected theoretical values, and small deviations can mostly be attributed to trace lengths that deviate from the theoretically assumed 200 mm. For a 1.8 m probe, the total resistances for the GND, VCC, SDA, SCL, and CLK traces are on average 2.046 Ω, 4.367 Ω, 4.307 Ω, 4.436 Ω, and 4.269 Ω, respectively. These results indicate the benefit of maximizing the GND trace width: this trace forms a return path for all signals, so a minimal GND resistance results in a smaller voltage drop for each signal of the probe. Figure 6 can be used to determine the maximum supply current of 7.15 mA along a 1.8 m probe. This results in a maximal supply voltage drop of 45.8 mV, which is not expected to affect sensor operation when using a >3 V battery supply.

#### 3.3.2. Thermal Effects and Epoxying

Figure 10 presents the change in contact resistance Δ*R* over a 200 mm probe section, as a result of epoxying, freezing, or heating the probe. The epoxy process does not seem to affect the contact resistance of the connectors, as all measured changes in resistance are close to 0 Ω, as theoretically expected. The freezing process clearly affects the trace resistance of a probe. Figure 10 depicts the theoretical trace resistance at −23${\;}^{{}^{\circ}}$C and 0${\;}^{{}^{\circ}}$C, calculated using $\rho_{-23}\;=\;1.387\;\times \;10^{-8}\;$Ωm and $\rho_{0}\;=\;1.543\;\times \;10^{-8}\;$Ωm [51]. The measurements are centered around the 0${\;}^{{}^{\circ}}$C trace resistance, which makes sense given the defrosting process. After equilibration to room temperature, all results return to close to their original values. The results of the heating experiment show increases of the trace resistance by up to 33 mΩ. Given the extreme deformation of the probe due to plastic softening and swelling during this experiment, these results are considered unexpectedly positive. For comparison, the used JST SH connector has a reported 40 mΩ contact resistance after environmental testing [46]. Overall, no connector failures were experienced during any of these tests, and the heated probe still worked one year after the experiment.

#### 3.3.3. Probe Bending

Figure 11 shows the connector’s performance as a function of probe bending. The average change in resistance is measured for each signal over a 200 mm probe section, as a function of bending radius. This test was planned to be destructive, determining the point of failure. The results show that no noticeable change in resistance is observed for a bending radius up to 390 mm. At 290 mm bending radius, some significant changes in contact resistance are observed, possibly due to settling contacts. The probe still performs well under a 200 mm bending radius, but under a 150 mm radius an average contact resistance of 8.333 kΩ is observed. This change became permanent after the experiment and represents the maximum measurement range for our setup (50 kΩ/6), so we can assume that one of the sensor boards snapped under this extreme bending. We can conclude from our tests that we are able to bend and reliably use the probe up to a bending radius of 200 mm, far beyond the expected realistic use case scenario.

#### 3.3.4. Trace Resistance over Time

Figure 12 presents the change in trace resistance per 200 mm section over time. The results are obtained 1, 17, 66, and 352 days after assembly, respectively, allowing us to observe effects such as oxidation and creep in the connectors. The measurements indicate a spread of up to 46 mΩ and the daily average change in resistance is mostly negative, which could indicate that contacts slightly improve over time, but it could also be a result of measurement inaccuracies due to variations in test lead tip contact resistance. The results of this experiment demonstrate the long-term stability of the developed connectors. No failures or negative trends due to oxidation or creep were detected. The protective properties of epoxy and the electroless nickel/immersion gold plating on the boards are expected to contribute to these positive results.

### 3.4. Accuracy Assessment

The probe’s accuracy is assessed by studying the deformation errors ($\epsilon_{d}$) for k∈{1,2,4,8,16,32} under 0${\;}^{{}^{\circ}}$C, ±1${\;}^{{}^{\circ}}$C and ±10${\;}^{{}^{\circ}}$C temperature variations. Figure 13 shows the 95% confidence interval (2.5 and 97.5 percentiles) of $\epsilon_{d}$ deformation errors over a 1 m probe. For each scenario (*k*, temperature) the results were obtained through a series of 10,000 Monte Carlo simulations, as explained in Section 2.5. Increasing the number of averaged measurements has a significant impact on the accuracy up to k=8, reducing the 95th percentile error range from ±1.92 mm to ±0.82 mm. Thereafter, the error range decreases more slowly to ±0.73 mm for k=32. Temperature variations lead to increased errors in deformation measurements. A variation of ±1 ${}^{{}^{\circ}}$C increases the 95% certainty interval up to 0.72 mm, while a variation of $\pm 10{\;}^{{}^{\circ}}$C widens the interval up to 2.88 mm. However, higher values for *k* still significantly improve the probe’s accuracy, regardless of the temperature variations. Given the overall superior results and limited impact on battery life for higher values of *k*, we advise the use of k=8–32 for all future measurements with the developed deformation probes.

### 3.5. Field Experiment

Figure 14 shows the data acquired by a probe (k=32) deployed on the Seward Peninsula, AK in September 2021. For this probe, data from the first five days after installation showed movements of up to 0.01 m, with a clear slip plane at 1.1 m depth related to the interface between frozen and unfrozen conditions. The data show continuous deformation with almost constant deformation rates of about 2 mm/day between 0 and 0.6 m depth along the *x*-axis. This represents a significant displacement since it exceeds the 95% confidence interval for k=32 at $\pm 1{\;}^{{}^{\circ}}$C of ±1 mm/m. Below this, deformation rates decrease until 1.1 m depth, and deformations below that are negligible. Similarly, no distinct movement can be observed along the *y*-axis for depths below 1.1 m, whereas above movements of up to 5 mm over this five-day period were recorded. The temperature data show that below 1.1 m depth the soil is frozen, whereas above the soil is unfrozen.

This clear deformation profile, with five sensors showing no movement and 11 sensors showing continuous deformation (both along the *x* and *y* direction), highlights the accuracy and real-world applicability of this approach. From the data, we can show that deformations in the order of 1 mm can be retrieved (even though this is below the nominal accuracy), and that the sensors themselves, when installed in a soil column, show no considerable drift or noise over the five-day period. Similar profiles have been observed at different locations with similar temperature profiles, while no deformation was recorded in areas characterized by the absence of permafrost. The inclusion of temperature measurements clearly indicates a relationship between freezing depth and soil movement, and will be the target for future investigations to better understand and predict soil transport in permafrost environments.

## 4. Conclusions

This study presents the design of a novel, low-power, flexible sensor array for monitoring soil deformation and temperature in slopes with shallow instabilities. In contrast with conventional approaches, the developed solution is low-cost, lightweight, robust, and easy to install, enabling large-scale deployments in densely distributed, wirelessly connected configurations. We provide a theoretical approach to deformation sensing based on three-dimensional acceleration measurements. This results in a conceptual probe design consisting of an array of temperature sensors and accelerometers spaced 50 mm apart. In a discussion of the electronic design, we present a configuration of individually addressable sensors on an I${}^{2}$C bus. A discrete shift register with D flip-flops along the entire probe enables cascaded sensor setups of variable length by using just six signals. The ultra-low-power design is compatible with an existing 2× AA battery-powered wireless data logger. A study of the sensor probe’s power consumption compares different probe lengths and measurement averagings (*k*), indicating that a worst-case scenario of a 1.8 m long probe with k=32 still results in a theoretical battery life of almost 12 years (not taking self-discharge into account), with the probe only representing 29% of the device’s total energy consumption. In order to meet the challenging mechanical requirements of the sensor probe, a specific electromechanical design is presented, using narrow FR4 printed circuit boards with a novel, solderless board-to-board connection method. The low-cost connection method does not require any components and enables extremely thin, semiflexible probes of adjustable length. In the final assembly step, the sensor arrays are placed in a 9.5 mm OD tube that is filled with epoxy compound. Given the novelty of the proposed interconnection method, we include an extensive study of the contact resistance and its change as a function of time, temperature, and deformation. The results demonstrate long-term stability and resistance to bending up to a radius of 200 mm. For the entire probe assembly, we evaluate the flexibility, showing significant deformation under small (<1 N) forces and nearly isotropic behavior. This demonstrates that the probe’s deformation is representative for soil movement. We also provide an assessment of measurement accuracy, showing that deformation measurements with k=32 and a constant temperature have a 95% confidence interval of ±0.73 mm/m. Under $\pm 10{\;}^{{}^{\circ}}$C temperature variations, this increases to ±2.17 mm/m. A set of probes was installed as part of a field experiment in a permafrost environment. The results show continuous soil displacement at a rate of 2 mm/day starting from the interface between frozen and unfrozen soil. This example emphasizes the importance of linked temperature and deformation measurements. Future work will focus on the installation of dense networks of distributed sensor arrays, wirelessly collecting temperature and deformation data in real time as part of an early warning system for slope instabilities. We will also investigate the use of this sensor system in different application domains. Seismic monitoring and instant response systems, for e.g., debris flows, can be conceived without changes to the probe’s hardware by implementing a continuous monitoring approach. For applications that require probe lengths of tens of meters, a hardware design with a differential I${}^{2}$C bus can be developed.

## 5. Patents

The solderless connector solution that is presented in this paper has been filed as U.S. Patent Application serial No. 17/543032.

## Figures and Tables

**Figure 1 sensors-22-02814-f001:**
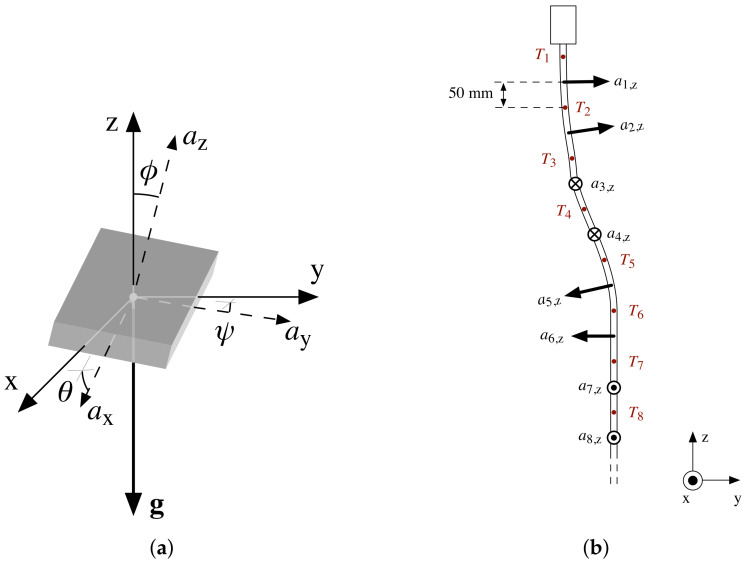
Accelerometer arrays can be used for deformation measurements. (**a**) Inclination measurement with an accelerometer and (**b**) design of the probe with temperature sensors ($T_{i}$) and accelerometers ($a_{i}$).

**Figure 2 sensors-22-02814-f002:**
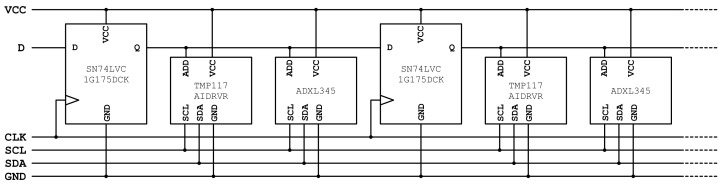
Schematic of cascaded temperature sensors and accelerometers in a shift register configuration.

**Figure 3 sensors-22-02814-f003:**
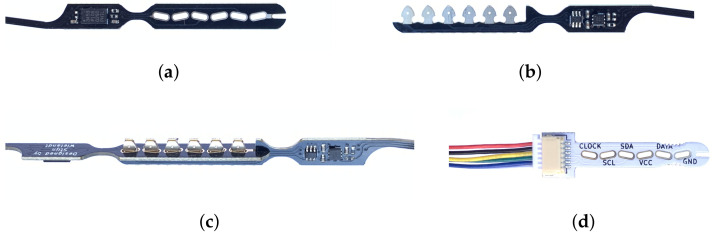
Close-ups of the proposed connector technology. (**a**) Female connector piece; (**b**) Male connector piece; (**c**) Mated connector; (**d**) Wire-to-Board adapter.

**Figure 4 sensors-22-02814-f004:**
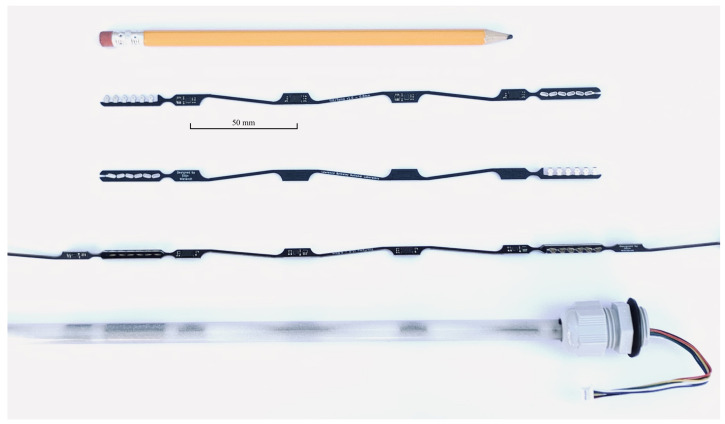
Modular design of the sensor probe. From top to bottom: top view of a sensor board, bottom view of a sensor board, cascaded sensor boards, and final probe assembly containing an array of cascaded sensor sections in a 3/8 in. (∼10mm) outer diameter tube, filled with epoxy.

**Figure 5 sensors-22-02814-f005:**
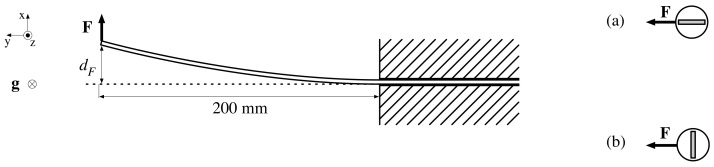
Test setup to measure the deformation $d_{F}$ of a 20 cm long probe segment as a result of a sideways force F, which is applied (**a**) in the PCB plane, or (**b**) perpendicular to the PCB plane.

**Figure 6 sensors-22-02814-f006:**
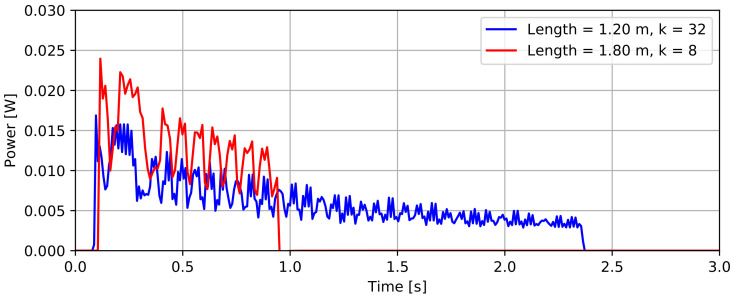
Power profiles for a 1.20 m long probe with 32 averaged measurements and a 1.80 m long probe with 8 averaged measurements.

**Figure 7 sensors-22-02814-f007:**
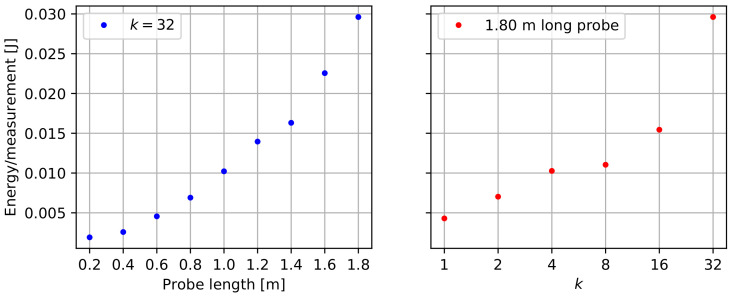
Energy per measurement as a function of probe length or the number of averaged measurements (*k*).

**Figure 8 sensors-22-02814-f008:**
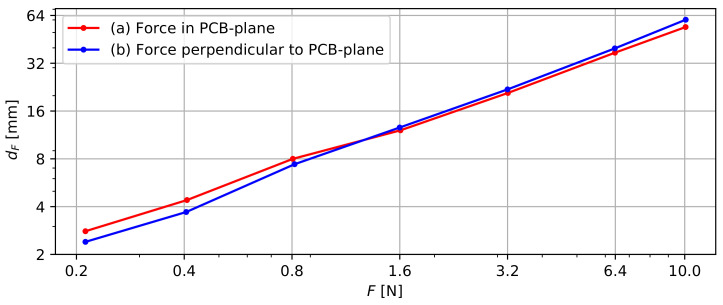
Deformation $d_{F}$ of a 200 mm long probe segment as a result of a sideways force F, either (a) in the PCB plane or (b) perpendicular to the PCB plane.

**Figure 9 sensors-22-02814-f009:**
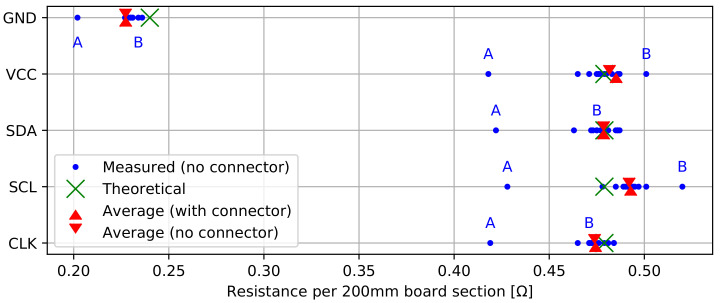
Trace resistance per 200 mm probe section, with and without connectors. Outlying trace resistance values for the same board are labeled as ‘A’ or ‘B’.

**Figure 10 sensors-22-02814-f010:**
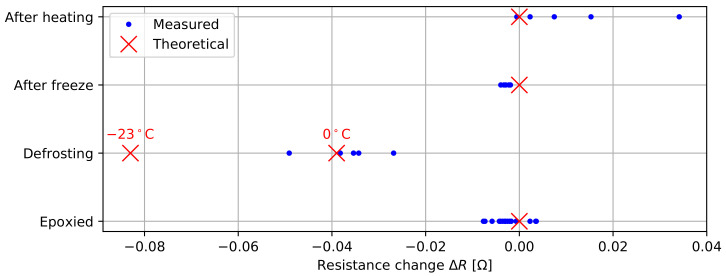
Trace resistance under freezing, heating, and epoxying.

**Figure 11 sensors-22-02814-f011:**
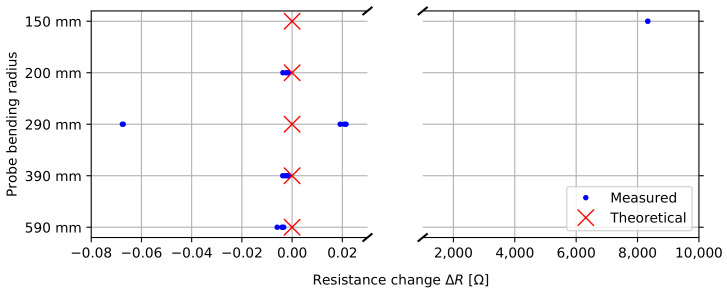
Trace resistance under bending.

**Figure 12 sensors-22-02814-f012:**
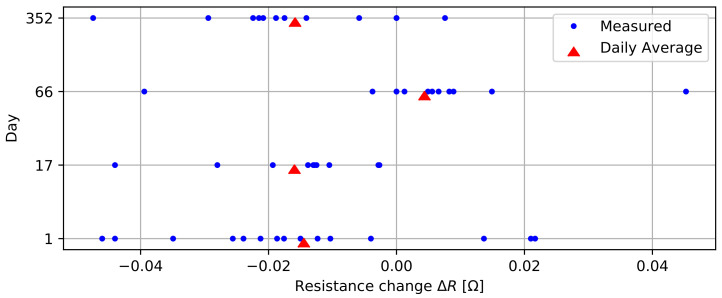
Trace resistance over time.

**Figure 13 sensors-22-02814-f013:**
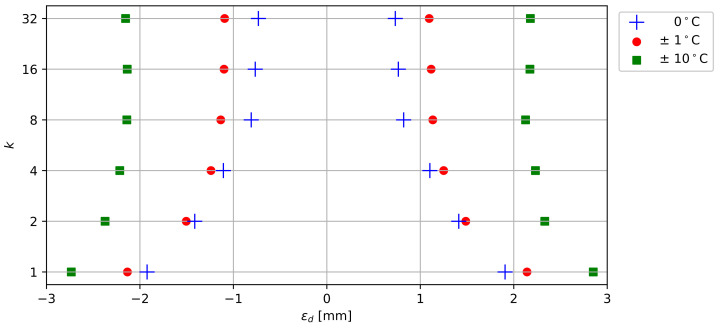
Deformation error ($\epsilon_{d}$) in function of number of averaged measurements (*k*) expressed as 2.5th and 97.5th percentiles for 0${\;}^{{}^{\circ}}$C, ±1${\;}^{{}^{\circ}}$C and ±10${\;}^{{}^{\circ}}$C temperature variations.

**Figure 14 sensors-22-02814-f014:**
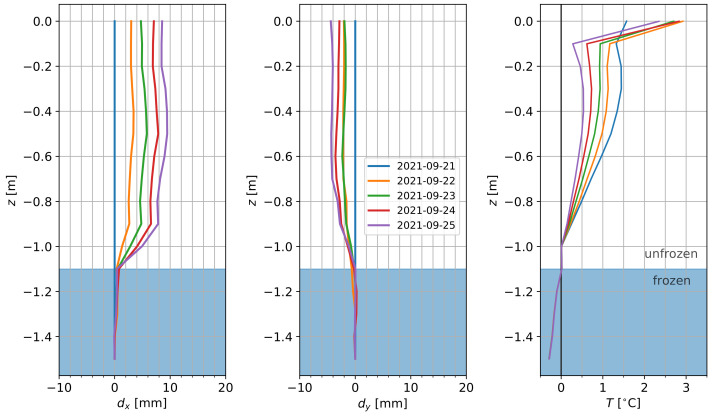
Field measurements obtained over five days at a field site on the Seward Peninsula, AK which is experiencing soil movements in response to permafrost thaw. Left and middle panels show the displacement along the *x* and *y*-axis, respectively; right column shows the soil temperature. Highlighted in blue is the frozen part of the subsurface, showing that the slip plane is colocated with the interface between frozen and unfrozen soil.

## Data Availability

The data presented in this study are openly available in the Next Generation Ecosystem Experiments Arctic Data Collection at https://doi.org/10.5440/1860494.

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
