# Peer review of "Low-Power, Flexible Sensor Arrays with Solderless Board-to-Board Connectors for Monitoring Soil Deformation and Temperature"

_sensors, 2022, doi:10.3390/s22072814_

Round 1

Reviewer 1 Report

This paper presents a instrument with accelerometer and temperature sensors for monitoring soil deformation and warning subsurface landslide. The paper is well organized with introduction, methods and results. Everything I except is discussed, including error analysis, thermal expansion of epoxy, sensor align calibration, etc. There are only a few comments:

  1. the authors discussed about aligning the sensors with the bottom one within one tube. Due to it is a WSN, is it necessary to align or at least aware of the orientation of each tube?
  2. In the results, only one tube data is shown? but there were 60 tubes installed. Would be nice to see the data as a sensor array.
  3. The temperature varies temporally, can it still be linked to deformation or hydrological parameters?
  4. More information about wireless network is suggested. Is the data transferred to a station in real time? How far away does the wireless network cover? Or the data is stored in the MCU and transferred by crew walking close to it with a wireless receiver?
  5. The authors use 50 Hz data rate, I think 100 Hz may be better. a) the datasheet shows 100Hz ODR has similar noise level as at 50Hz. a) 50Hz is synchronized with powerline frequency in some countries, it may introduce noise when averaging. While powerline harmonics may be cancelled with 100 Hz ODR.

Some suggestion about the instrument:

  1.  it may be powered with a tiny solar panel.
  2. apply differential IIC bus to expand its length up to 20 m.

Author Response

Response to Reviewer 1 Comments

The authors would like to thank you for your thorough review with on-point questions, remarks, and suggestions. We appreciate your input and we believe it has improved our paper.

1) The authors discussed about aligning the sensors with the bottom one within one tube. Due to it is a WSN, is it necessary to align or at least aware of the orientation of each tube?

When probes are deployed, the orientation of each tube can be recorded. However, this is usually not necessary for landslide monitoring, since the deformation will occur in the direction of the slope. We have added this to the paper.

2) In the results, only one tube data is shown? but there were 60 tubes installed. Would be nice to see the data as a sensor array.

Probes that were installed in similar temperature regimes showed very similar deformation profiles. Hence, the one shown serves as an example for a typical deformation profile. Other probes that were installed in environments with no permafrost showed no deformation over the 5 day period. Here, we are focusing on the development of the probe and not the application. The distributed deformation data will be the focus of a following publication, that will look at more than 5 days of data. We clarified this in the text.

3) The temperature varies temporally, can it still be linked to deformation or hydrological parameters?

The temporal dynamics of ground temperature are actually used to estimate the hydrological parameters, and are potentially important in permafrost environments to trigger soil instabilities. So yes, the temporal dynamics are perhaps even more important than just a single measurement in time. We elaborated on this in the revised version.

4) More information about wireless network is suggested. Is the data transferred to a station in real time? How far away does the wireless network cover? Or the data is stored in the MCU and transferred by crew walking close to it with a wireless receiver?

The data logger provides Bluetooth Low Energy connectivity for local data transfers in the field, and long-range connectivity for real-time streaming through LoRa. We provided more information on these wireless networking provisions in Section 2.3.1 and 2.5

5) The authors use 50 Hz data rate, I think 100 Hz may be better. a) the datasheet shows 100Hz ODR has similar noise level as at 50Hz. a) 50Hz is synchronized with powerline frequency in some countries, it may introduce noise when averaging. While powerline harmonics may be cancelled with 100 Hz ODR.

Thank you for your input. We agree that the noise performance under 50 Hz is similar to the 100 Hz noise level and our phrasing was confusing. Our choice for the 50 Hz data rate was motivated by the lower supply current and lower allan deviation. Furthermore, the 50 Hz ODR results in a measurement bandwidth (-3dB point) of 25 Hz, reducing the sensitivity to powerline frequencies. We did not conduct specific EMC tests, but our lab experiments did not reveal any issues that could be associated with powerline interference.

Some suggestion about the instrument:

1) it may be powered with a tiny solar panel.

Thank you for this suggestion, we added this idea to the power discussion in the paper.

2) apply differential IIC bus to expand its length up to 20 m.

We included this suggestion under future work.

Reviewer 2 Report

A new monitoring system applicable in landsliding assessment was presented in the present study with detailed technical descriptions of the sensor development and preliminary “in-situ” results. As the monitoring of landslides is a very important area in geotechnical engineering and geosciences, but with certain challenges in the implementation, the present work provided a new contribution in this area of research (and practice) with promising application of the newly developed system. I would have some comments to the authors which I hope can be found useful in a potential revision. Overall, I found the work of high-quality, though I would like to check with the authors on some possible limitations with respect to the frequencies detected as these sensors are in reality measuring acceleration in the first place. Please see my detailed comments below.

(1) Measurement of temperatures at deeper ground has its own value in many geotechnical engineering applications, however slope instability might be often related with near-surface processes. Please elaborate more on the relevance of the temperature measurements in association with instrumentation for slope stability monitoring and whether the intention is the temperatures to be measured near-surface or deeper.

(2) Some distinction of landslide types could be briefly discussed in the Introduction, particularly in association with instrumentation and monitoring aspects. Landsliding associated with tailing dams is also one of the important threats to the environment which can be mentioned in the Introduction section.

(3) Lines 60-63: Some potential limitations with wireless monitoring could be briefly discussed.

(4) Accelerometer usage could have some potential limitations especially in “early-warning” assessment of potential landsliding?

(5) Relative advantages of the present proposed monitoring technique compared with fiber sensing could be elaborated.

(6) Lines 121-131: Monitoring of landslides may often be in the vicinity of railway or highway systems where vibrations might be significant. This can be a very common case in high density urban areas in hilly terrain, as example. How this technique can be implemented on sites with potential important vibration noise? Please provide some comments to this.

(7) Would there be some limitations especially in low frequency measurements with this array system?

(8) Lines 228-235: These descriptions imply major application in “continuous” landsliding rather than types such as debris flows. Please explain whether the method could have additional applications (or based on future modifications) to debris flow monitoring.

(9) Please explain if filtering has been applied to the results in figure 6 or these are raw data.

(10) References style: year could be removed from the references cited within the main text (based on journal’s general style as it adopts numeric-style). The year can appear in the reference list at the bottom of the manuscript.

(11) If I understood correctly, figure 13 data show the sensitivity to temperature changes (not the influence of the absolute temperature). Please provide some clarifications to this along with some brief discussion (similar to a previous comment of mine) on the importance of temperature measurements (or changes in temperature) in relation to landsliding.

Author Response

Response to Reviewer 2 Comments

The authors would like to thank you for your thorough review with on-point questions, remarks, and suggestions. We appreciate your input and we believe it has improved the quality of our paper.

(1) Measurement of temperatures at deeper ground has its own value in many geotechnical engineering applications, however slope instability might be often related with near-surface processes. Please elaborate more on the relevance of the temperature measurements in association with instrumentation for slope stability monitoring and whether the intention is the temperatures to be measured near-surface or deeper.

We added more detail on this in the introduction. Recent work has shown that distributed temperature measurements in the shallow subsurface can be used to estimate variations in soil moisture. Such data will be very useful in linking slope instabilities with their triggering factors. Similarly, permafrost thaw is known to cause slope instabilities and hence having a good understanding of the soil temperature may help in linking those processes and using it for early warning. Here we are just focusing on these near-surface probes, hence the temperature dynamics we are referring to are also the ones observed in the upper 1.2 to 1.8 m depth. 

(2) Some distinction of landslide types could be briefly discussed in the Introduction, particularly in association with instrumentation and monitoring aspects. Landsliding associated with tailing dams is also one of the important threats to the environment which can be mentioned in the Introduction section.

That is a good point. We did not specify this in the introduction, but it was focused on shallow, rainfall induced landslides in both natural and engineered slopes. We have added a short paragraph in the introduction to clarify this. 

(3) Lines 60-63: Some potential limitations with wireless monitoring could be briefly discussed.

We elaborated more on the limitations of wireless sensor networks for monitoring deformation dynamics, as well as the challenges of networked sensor deployments in the field.

(4) Accelerometer usage could have some potential limitations especially in “early-warning” assessment of potential landsliding?

We use accelerometers to regularly measure the tilt of each sensor, but the sensors are not recording continuously. This means that they regularly check for a change in static conditions, but they are not appropriate to detect early signs of, e.g., rock falls. We clarified these limitations in the paper.

(5) Relative advantages of the present proposed monitoring technique compared with fiber sensing could be elaborated.

We clarified the advantages of our proposed technique w.r.t. the shortcomings of fiber optic and remote sensing solutions, as presented in the introduction.

(6) Lines 121-131: Monitoring of landslides may often be in the vicinity of railway or highway systems where vibrations might be significant. This can be a very common case in high density urban areas in hilly terrain, as example. How this technique can be implemented on sites with potential important vibration noise? Please provide some comments to this.
We have now addressed this in the paper: “In infrastructure monitoring scenarios (e.g., railways, highways), vibrations might be more prevalent. In this case, sensor noise can be reduced by averaging measurements over time, and lowering the sensor's measurement bandwidth below the vibrational frequencies.” We have also specified the measurement bandwidth for our configuration (25Hz) in the paper.

(7)   Would there be some limitations especially in low frequency measurements with this array system?

Here, the intention is to use acceleration to extract displacement and not monitor the ambient seismicity. We purposefully designed the system to be low-power, regularly handling small amounts of data. We clarified this in Section 2.1.

(8)   Lines 228-235: These descriptions imply major application in “continuous” landsliding rather than types such as debris flows. Please explain whether the method could have additional applications (or based on future modifications) to debris flow monitoring.

Our system performs regular deformation measurements, rather than continuously monitoring movements. We have made this more clear in this paper revision. As such, the system is targeted at shallow slow-moving landslides rather than rapid debris flows. However, the system could possibly detect the triggering of a debris flow. We have added additional applications to our future work discussion in Section 4.

(9) Please explain if filtering has been applied to the results in figure 6 or these are raw data.

The current measurements were performed with an integration interval of 1.16 ms and a 10X averaging filter, resulting in a total data acquisition rate of 86 samples per second. We added these data acquisition parameters to the paper.

(10) References style: year could be removed from the references cited within the main text (based on journal’s general style as it adopts numeric-style). The year can appear in the reference list at the bottom of the manuscript.

We have changed the references to follow the journal’s style guide.

(11) If I understood correctly, figure 13 data show the sensitivity to temperature changes (not the influence of the absolute temperature). Please provide some clarifications to this along with some brief discussion (similar to a previous comment of mine) on the importance of temperature measurements (or changes in temperature) in relation to landsliding.

In the introduction, we provided a discussion on the importance of temperature measurements, following your comment (1). In Section 2.1, we clarified the impact of temperature changes on accelerometer drift, and the effect of sensor offset, which is linked to the absolute temperature:

“Sensor drift can happen over time and under influence of changes in temperature and/or supply voltage. For subsurface applications, temperature changes are usually limited and slow. Furthermore, drift can be detected, compensated, and prevented by stabilizing the supply voltage and by performing measurements of temperature and battery voltage. Offset errors (due to the absolute temperature or the sensor's manufacturing process) are usually eliminated through a calibration process. Deformation measurements, however, are not affected by offsets and do not require calibration, since only changes in inclination are measured.”

In Section 2.5, we clarify the importance of measuring temperature changes:

“The temperature sensors are used to characterize shallow subsurface processes (e.g., permafrost thaw, variations in soil moisture), but temperature values can also provide insights in the accuracy of deformation measurements. As stated by the manufacturer, temperature changes can induce a 0~g offset of acceleration measurements.”